# The role of structural connectivity on brain function through a Markov model of signal transmission

Rostam M. Razban[1]*, Anupam Banerjee[1], Lilianne R. Mujica-Parodi[1,2], Ivet Bahar[1,3]*

1 Laufer Center for Physical and Quantitative Biology, Stony Brook University, Stony Brook. New York, United States of America, 2 Department of Biomedical Engineering, Stony Brook University, Stony Brook, New York, United States of America, 3 Department of Biochemistry and Cell Biology, Renaissance School of Medicine, Stony Brook University, Stony Brook,New York, United States of America,

* rmr4442@gmail.com (RMR); bahar@laufercenter.org (IB)

## Abstract

Structure determines function. However, this universal theme in biology has been surprisingly difficult to observe in human brain neuroimaging data. Here, we link structure to function by hypothesizing that brain signals propagate as a Markovian process on an underlying structure. We focus on a metric called commute time: the average number of steps for a random walker to go from region A to B and then back to A. Commute times based on white matter tracts from diffusion MRI exhibit an average±standard deviation Spearman correlation of −0.26±0.08 with functional MRI connectivity data across 434 UK Biobank individuals and −0.24±0.06 across 400 HCP Young Adult brain scans. The correlation increases to −0.36±0.14 and to −0.32±0.12 when the principal contributions of both commute time and functional connectivity are compared for both datasets. The correlations are stronger by 33% compared to broadly used communication measures such as search information and communicability. The difference further widens to a factor of 5 when commute times are correlated to the principal mode of functional connectivity from its eigenvalue decomposition. Overall, the study points to the utility of commute time to account for the role of polysynaptic (indirect) connectivity underlying brain function by assuming that signals randomly traverse along the underlying brain structure.

## Introduction

The brain is a network of neuronal regions (nodes) connected by white matter tracts (edges) that are stochastically organized for effective signaling [1]. When tracts between regions are improperly constructed or removed, neuronal signaling will be hampered. This can lead to impaired function and thus neurological diseases [2–10]. Yet, despite the important role of brain network connectivity, it remains unclear to what extent the brain's structure constrains its function [11,12]. Toward this goal,

**Data availability statement:** Scripts necessary to reproduce figures can be found at github.com/rrazban/brain_commute. Processed dMRI and fMRI data used by the scripts for the HCP Young Adult dataset can also be found on GitHub at github.com/rrazban/brain_commute/tree/main/data/hcp_ya_100. Please refer to the respective UK Biobank (ukbiobank.ac.uk) and HCP Young Adult website (db.humanconnectome.org) to access previously published preprocessed dMRI and fMRI images (47, 91).

**Funding:** Support from the National Institutes of Health (R01 DK116780 and R01 DA062680) is gratefully acknowledged by IB. The funders had no role in study design, data collection and analysis, decision to publish, or preparation of the manuscript.

**Competing interests:** The authors have declared that no competing interests exist.

connectivity matrices derived from diffusion MRI (dMRI) have been compared to resting-state functional MRI (fMRI) data. dMRI can measure the number of white matter tracts between brain regions [13], and describes the connectome. fMRI, on the other hand, measures time-series of regional activity, which reflects region-dependent brain function [14]. Studies directly comparing dMRI and fMRI data across individuals have found surprisingly low correlations between the two sets [15–17], raising questions about the extent of dependency of brain function on the connectome.

The weak relationship between structural and functional connectivity can be attributed to the two MRI-derived modalities measuring different aspects of anatomical connectivity. The fact that two brain regions are not directly connected by a white matter tract, does not imply a lack of functional connectivity. The path of tracts through intermediate regions connecting the two brain regions of interest can influence their functional connectivity strength. Polysynaptic (indirect) connectivity goes beyond an edge-centric picture by accounting for the entire structural network of the brain [11,18].

Metrics under the class of communication models have been developed to better capture polysynaptic connectivity [19–22]. Inspired by recent results employing communication measures, we investigate whether a so far unexplored metric, commute time, could better link structure to function. Commute time is a global property, that depends on the entire network topology representing the connectome, rather than just a property of a node pair. As such, it might provide better insights into functional consequences of the connectome. Already several metrics exist for capturing polysynaptic connectivity [18]. For example, communicability also captures polysynaptic connectivity by considering diffusion processes. However, the corresponding physical constraints implied by its mathematical expression do not have a clear relationship to signaling dynamics (Methods, Equation 13). Here, we apply a physically motivated property with the objective of yielding further insights into the structure-function relationship of the human brain as measured by dMRI and fMRI.

Commute time is the average number of steps to randomly diffuse back and forth between two nodes in a network. Applying commute time physically postulates that signaling propagates as a Markovian process on an underlying structure [23–26]. Signals traverse from one brain region to another based on the relative number of white matter tracts connecting the two regions relative to all other tracts emanating from the currently located region. Commute time is a biologically reasonable first approximation for brain signaling dynamics because it assumes that traversing signals have no global information about the brain's overall structure. Signaling is constrained by the local structure sequentially seen at each of the brain regions that the signal passes on its way to its final destination.

An analytical expression for commute time can be derived, enabling its quick calculation from the structural connectivity matrix, as outlined in the next section. A few studies have already applied commute time specifically to fMRI [27,28]. We extend those studies in the Results by calculating the commute time based on brain structure as derived from dMRI to investigate to what extent structural connectivity affects function as derived from fMRI (Fig 1).

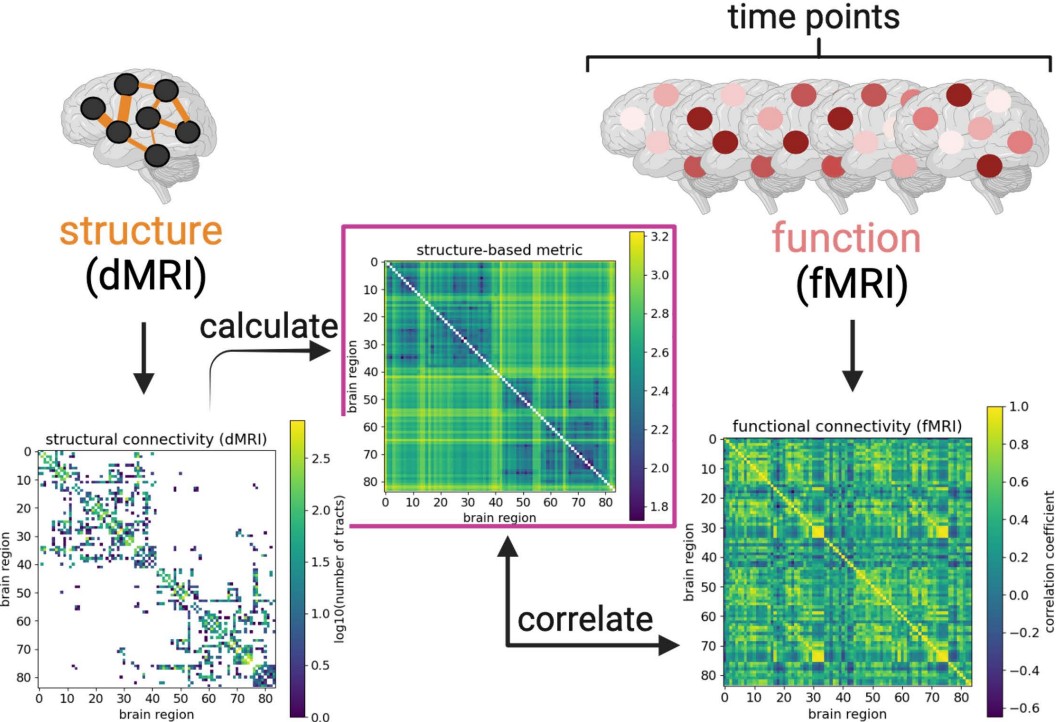

**Fig 1. Bridging between brain structure and function by calculating commute time.** Connectivity matrices derived from diffusion MRI and functional MRI data are illustrated from an arbitrarily chosen UK Biobank individual (subject id: 1000366). Brighter colors reflect larger values (denser white matter tracts, larger commute times, or larger correlation coefficients), while structural matrix elements equal to 0 (no white matter tracts) before log transformation are colored white. Rows and columns refer to 84 brain regions from the Desikan-Killiany atlas [29,30]. Brain regions are labeled such that the first half corresponds to the left hemisphere; second half, right hemisphere. The structural connectivity matrix (left map) yields the commute time matrix in the middle (Methods), which is then compared with correlations calculated from the fMRI time-series (right map). Figure created with biorender. com.

## Brain signaling as a Markov stochastic process

A Markovian stochastic process posits that state transitions are exclusively based on its current state; there is no memory of the past states [23,31,32]. For a brain signal traversing from region $i$ to region $j$, the Markov process depends on all possible sequential single steps from region $i$ that eventually reach region $j$, e.g., $i$ to $k$, $k$ to $l$, $l$ to $m$, ...., $z$ to $j$, each step being dependent on the instantaneous region. The Markovian metric we focus on in this paper is the commute time $C_{ij}$, which is defined as the average number of steps to go to from node $i$ to node $j$, and back to node $i$. This can be expressed in terms of the hitting time $H_{ij}$, the average number of steps to go from node $i$ to $j$,

$$C_{ij} = H_{ij} + H_{ji} \qquad (1)$$

Note that the hitting time is not a symmetric property, $H_{ij} \neq H_{ji}$, while the commute time is, $C_{ij} = C_{ji}$. Because structural and functional connection matrices derived from dMRI and fMRI are symmetric, we focus on $C_{ij}$, rather than $H_{ij}$.

To derive an analytical expression for $C_{ij}$, we first need an expression for $H_{ij}$. For a network of $N$ nodes, the hitting time $H_{ij}$ can be expressed in terms of the elements $M_{ik}$ of the Markov transition matrix, **M**, which describe the transition probability from node $i$ to $k$, followed by succeeding steps starting from $k$ using the recursive expression [23],

$$H_{ij} = \sum_{k}^{N} M_{ik} \left(1 + H_{kj}\right)$$

(2)

Because the probability of taking a step to another node $k$ only depends on the possible transitions from the current node $i$, Equation 2 describes a Markov process. $M$ can be expressed in terms of the adjacency matrix $A$ as $M = D^{-1}A$, where $D$ is a diagonal matrix, the elements of which are obtained by summing the corresponding row of $A$, i.e.,

$$D_{ii} = \sum_{k}^{N} A_{ik}$$

(3)

In this study, we use dMRI tractography data to construct a weighted adjacency matrix $A$. Thus, the element $A_{ij}$ is the number of streamline counts of white matter tracts between brain regions $i$ and $j$. Note that the weighted adjacency matrix is equivalent to the structural connectivity matrix. We use the words adjacency and structural connectivity interchangeably throughout the text. We further note that $A$ can be expressed in terms of the Laplacian or Kirchhoff matrix $\Gamma$,

$$\Gamma = D - A$$

(4)

As outlined in the Methods and previously derived [23], introducing the Laplacian matrix transforms the self-consistent equation in Eq. 2 with $H$ on both sides of the equality to a simple linear equation. Then, the hitting time for diffusing from node $i$ to node $j$ in an arbitrary network obeying Markovian stochastics can be expressed.

$$H_{ij} = \sum_{k}^{N} \left( \left[\Gamma^{-1}\right]_{ik} - \left[\Gamma^{-1}\right]_{ji} - \left[\Gamma^{-1}\right]_{jk} + \left[\Gamma^{-1}\right]_{jj} \right) D_{kk}$$

(5)

Inserting Equation 5 into 1, we obtain an analytical expression for commute time $C_{ij}$ [23,25].

$$C_{ij} = \left( \left[\Gamma^{-1}\right]_{ii} + \left[\Gamma^{-1}\right]_{jj} - 2\left[\Gamma^{-1}\right]_{ij} \right) \sum_{k}^{N} D_{kk}$$

(6)

As a validation of the above graph-theoretical approach for calculating commute times, we numerically calculate all possible $C_{ij}$ values for an arbitrarily chosen UK Biobank individual's dMRI-derived structural connectivity matrix. We find excellent agreement between theoretically (Eq. 6) and numerically calculated commute times across all brain region pairs (S1 Fig in SI). Thus, we proceed to repeatedly evaluate Equation 6 across all possible pairs of brain regions for a given individual using their dMRI scan to define the Laplacian matrix to investigate whether commute time can help bridge between structural and functional properties reflected by dMRI and fMRI, respectively.

## Results

### Commute time captures simulated function

We first aim to establish that commute times can in principle capture function reflected by correlations between time-series data, also called *functional connectivities* (*FCs*), among all pairs of brain regions. To this aim, we arbitrarily select an individual from the UK Biobank (subject id: 1000366). We process their dMRI data to construct the corresponding adjacency matrix $A$ representing the structural connectivity of white matter tracts under the Desikan-Killiany atlas (84 brain regions), where each matrix element corresponds to the number of tracts between the two respective brain regions

[29,30]. The latter is used to generate the Laplacian matrix (Eq. 4), for which we take the pseudoinverse to evaluate commute times (Eq. 6). We will compare these analytically evaluated commute times for all pairs of brain regions with the corresponding FCs inferred from simulations of brain signaling.

FCs in this subsection are generated by iteratively simulating a mean-field Ising system composed of 84 up or down spins with a single coupling strength parameter $\lambda$ (Methods, Fig 5). The underlying structure of the simulated Ising system is set to the same UK Biobank individual as in the previous paragraph. We choose a mean-field Ising system because of its simplicity (only one parameter which can be uniquely fitted to the data) and our previous results demonstrating its utility in capturing global brain dynamics [33,34]. Others have also applied different Ising model types to explain neuronal signaling in the brain [35–39]. The generated time-series across brain regions are corelated with each other to create the corresponding FC matrix. The comparison of commute time (Eq. 6) and simulated FC are presented in Fig 2A-B. In the Supplement, we corroborate our results with the Wilson-Cowan model [40,41], a more complicated (24 parameters, S1 Table in SI) and more realistic simulation of brain function (S2 Fig in SI). Both Wilson-Cowan and Ising models' outputs are convoluted with a hemodynamic response function [42] to output a signal similar to that of a fMRI blood oxygenation level-dependent (BOLD) signal.

Fig 2A demonstrates that dMRI-based commute times significantly capture functional connectivities from the Ising simulation for an arbitrarily chosen UK Biobank individual. The commute time-FC Spearman correlation coefficient ($\rho$) is −0.68 with a p-value (p) $< 10^{-300}$. To ensure that the strong correlation is a direct result of the underlying structure, we consider a null model where function is simulated on a shuffled structural connectivity matrix and now find essentially no correlation ($\rho$(commute time, FC) = −0.05, S3A Fig in SI). The correlation is negative because larger commute times mean smaller functional couplings as the two regions are separated by a larger number of intermediate steps, making their stochastic signaling less correlated. FC values may be influenced by many other factors besides structural connectivity. To potentially minimize noise and enrich its structural signature, we decompose the FC matrix into its principal components and compare its top two components to the theoretically predicted commute times in Fig 2B. The correlation becomes stronger ($\rho = -0.72$, $p < 10^{-300}$).

We further investigate how the structure-function relationship is affected by the coupling strength between brain regions, which is described by the Ising model's single parameter $\lambda$. When the coupling strength increases, a larger commute time-FC correlation is observed on average (S4 Fig in SI). Young adults' brains have been shown to have coupling strengths just below the Ising model's critical point that further decrease with age [34]. This roughly corresponds to the coupling strength value presented in Fig 2. Similar weakening in the commute time-FC relationship is seen for the Wilson-Cowan model when the coupling constant is decreased (S5 Fig in SI).

To ensure that our results are not an artifact of the particular individual's brain structure on which function is simulated, we present results for several individuals' brain structures in the Supplement (S6 Fig in SI). We also assess commute time's performance with respect to two leading communication metrics in the field, search information and communicability, and dMRI-derived connectivity as a control (Fig 2C). This analysis demonstrates that commute time yields a stronger structure-function correlation compared to all three metrics.

In summary, the above *in silico* tests demonstrate that the structure-based commute times between 84 brain regions and simulated FCs for the same brain region pairs have a strong correlation. Commute times are predicted by graph-theoretical analysis of connectome topology, using the number of white matter tracts as ingredients for the adjacency matrix, whereas functional connectivities are from simulations of signal dynamics on the same network of brain regions. The relatively strong correlation between the two sets confirms the utility of adopting commute time as a metric to explore to what extent a Markovian diffusion process on brain structure explains fMRI-derived FCs observed for all individuals in the UK Biobank, presented next.

## Commute time better captures brain function

With encouraging results from simulations of brain function, we proceed to compare the commute times to FCs from fMRI for the same individual, where both modalities are processed according to the Desikan-Killiany atlas (Methods). Fig 3A and B illustrate the results for the same arbitrarily selected UK Biobank individual in the previous subsection. A statistically

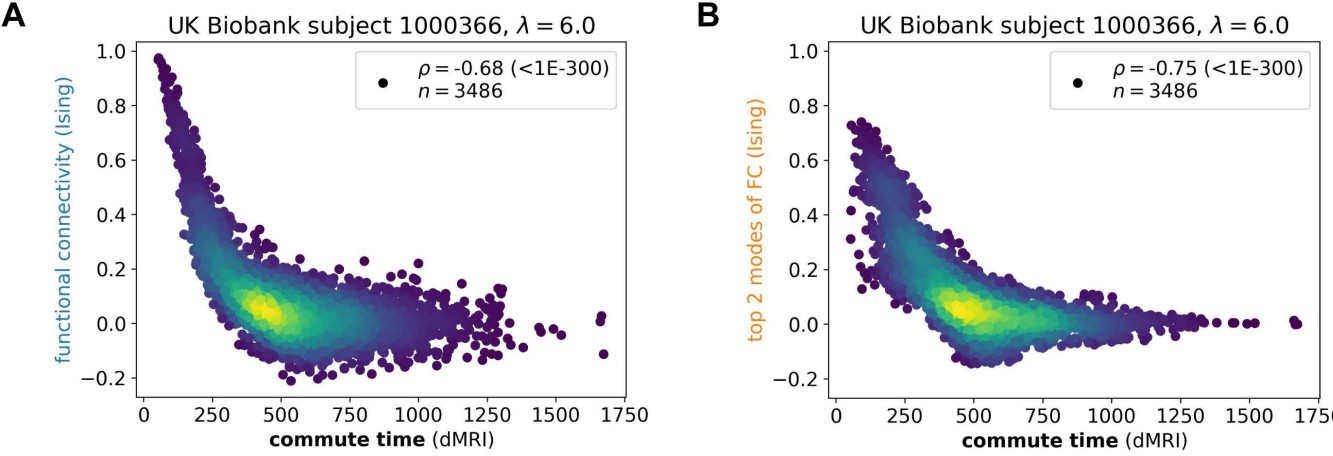

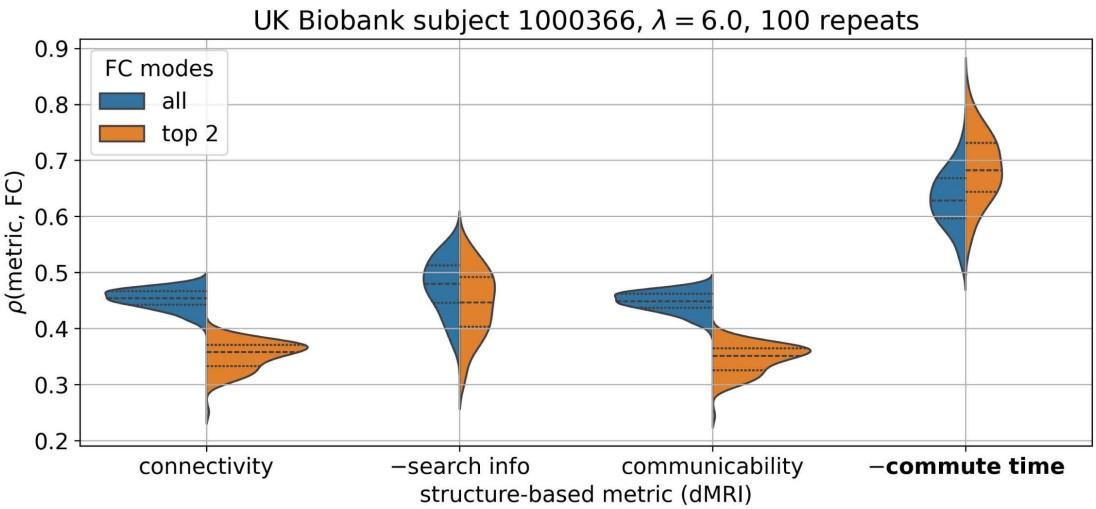

**Fig 2. Commute time captures functional connectivity in a mean-field Ising simulation of brain signaling.** The Ising simulation with a coupling strength $\lambda = 6.0$ is run on a dMRI-derived structure from an arbitrary UK Biobank individual (subject id: 1000366) under the Desikan-Killiany atlas (84 brain regions). **A**. Data points correspond to all possible pairs of brain regions (n = 84x83/2) and are colored to capture the density of points, with brighter colors corresponding to greater density. The ordinate represents the elements of the functional connectivity (FC) matrix obtained by an Ising simulation, and the abscissa shows the corresponding commute time based on graph theoretical analysis on dMRI data (Eq 6). **B**. Same as **A**, using only the top two principal components of the FC matrix. The symbol ρ corresponds to the Spearman correlation coefficient with its p-value following in parenthesis. **C**. Commute time exhibits a stronger correlation with simulated functional connectivity compared to three other metrics, connectivity, search information, and communicability. 100 replicate Ising runs are performed. For visualization purposes, the additive inverse (−) of search information and commute time are taken such that all structure-based metrics are positively correlated with FC. Each metric is described by two distributions: orange, based on the top two principal components of the FC matrix and blue, all components. Dashed lines within distributions correspond to quartiles: 25th percentile, 50th percentile (median) and 75% percentile. Corresponding Kolmogorov-Smirnov pairwise tests between metrics are summarized in S2 Table in SI.

significant (p = 4.0*10^{-44}) but weak correlation of $\rho = -0.23$ is observed across all possible pairs of brain regions. The correlation becomes stronger when considering FC's top principal component ($\rho = -0.28$, p = 7.8*10^{-63}). The second principal component is not included as in the Ising model, because the top eigenvalue already accounts for more than the sum of the first two of the Ising model (S7A Fig in SI).

The correlation between commute time and fMRI-derived FCs (Fig 3A-B) is much weaker than that observed between commute time and simulation-derived FCs (Fig 2A-B). One reason is that the deviation of the fMRI FC matrix from the dMRI structural connectivity matrix is much greater than that of the simulated FC from dMRI (S7B-C Fig in SI; Fig 1). Significantly larger functional couplings are observed between the two hemispheres in the fMRI data, compared to that inferred from Ising simulations, or that indicated directly by dMRI data. Inter-hemisphere couplings are significantly stronger in fMRI data, i.e., there is a fundamental qualitative gap between structural connectivity (from dMRI) and FC (from fMRI). We will return to this point in the Discussion by separately studying intra-hemispheric tracts and introducing tracts between the same brain regions across the two hemispheres.

Extension of our analysis to 434 UK Biobank subjects for which we have dMRI and fMRI data leads to an average ± standard deviation commute time-FC correlation of $-0.26 \pm 0.082$ (Fig 3C, *blue* distribution), which increases to $-0.30 \pm 0.11$ when only including the top principal component of FC (Fig 3C, *orange* distribution). To place these results into context, we compare them to dMRI-derived structural connectivity, search information and communicability [18]. We find that commute time yields a stronger structure-function correlation than the other three structural metrics (Fig 3C).

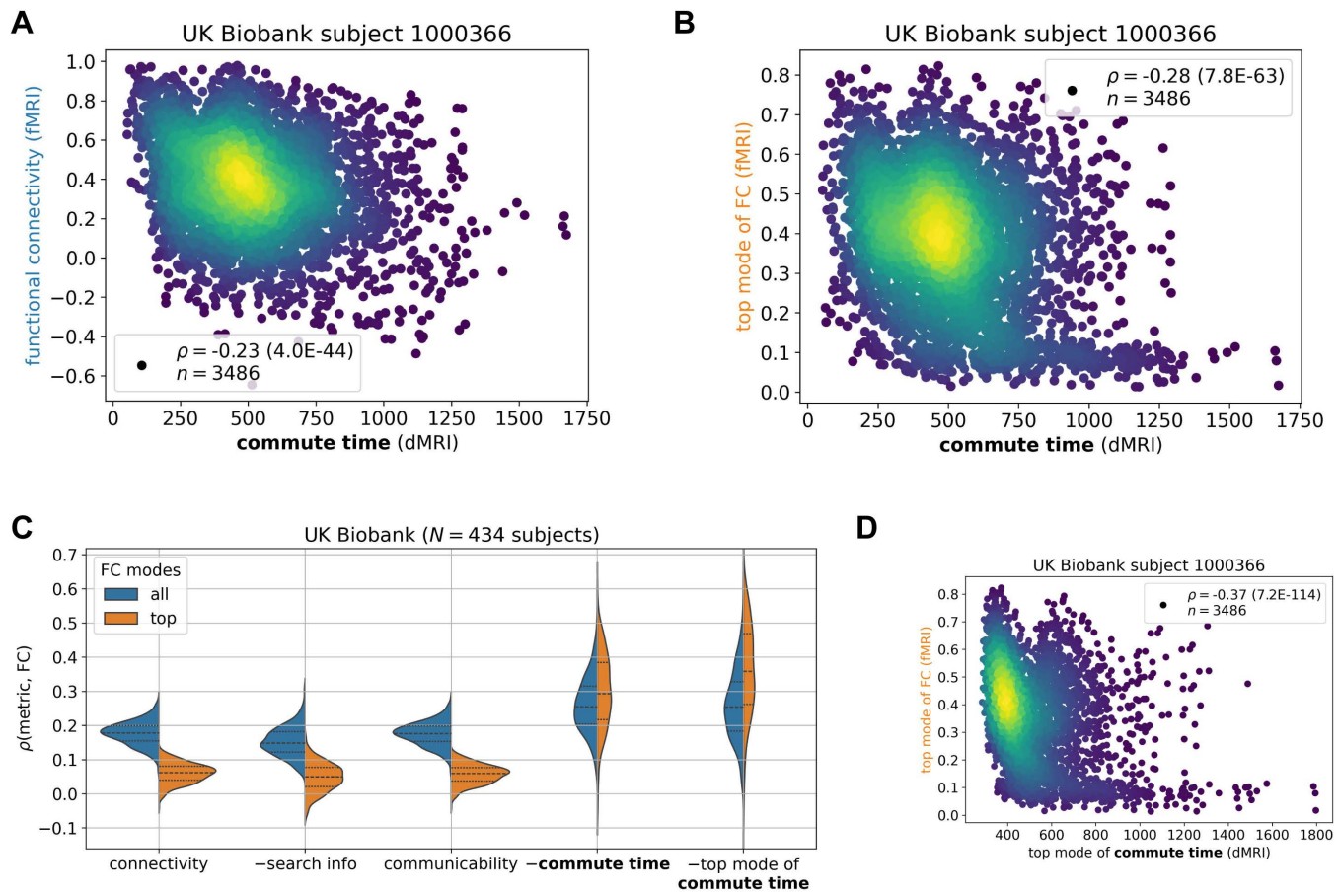

**Fig 3. Commute time marginally captures functional connectivity from functional MRI across the UK Biobank.** fMRI and dMRI data are processed according to the Desikan-Killiany atlas (84 brain regions). **A-B.** Calculated commute times (abscissa) are plotted against FCs indicated by fMRI data for one individual. See caption for Fig 2 for more details. **C.** Comparison of different metrics derived from structural connectivity. The ordinate shows their correlation to fMRI data. Corresponding Kolmogorov-Smirnov pairwise tests between metrics are summarized in S3 Table in SI. N corresponds to the number of human subjects considered from the UK Biobank. **D.** Values from the top mode of commute time (abscissa) are plotted against those from the top mode FC.

Moreover, when only including the top principal component of FC, the difference between the metrics widens (Kolmogorov-Smirnov test statistic (KS) = 0.59 vs 0.95 (S3 Table in SI)). This is because structural connectivity, search information and communicability predict FC's top mode more poorly than the entire FC across UK Biobank individuals. This hints that these metrics may be capturing nonstructural drivers of FC, since FC's top mode corresponds more strongly to structural constraints as supported by simulation results in the previous subsection.

To ensure that our results are not sensitive to the particular atlas choice for dMRI and fMRI processing, we generate results according to the finer Talairach atlas [43]. S8 Fig in SI shows that the commute time again outperforms other metrics; however, the correlations are lower across the board. This is consistent with previous results in the literature showing that atlases delineating more brain regions yield smaller structure-function correlations [15,44] (see [45] for an exception). Finally, to further corroborate our results, we examine dMRI and fMRI from the Human Connectome Project (HCP) Young Adult dataset [46,47]. We investigate the HCP Young Adult 100 Unrelated Subjects subset and find similar trends, $\rho = 0.24 \pm 0.06$ across 400 HCP Young Adult brain scans (S9A Fig in SI). To check whether results are dependent on the tractography method used to analyze the dMRI data, we perform probabilistic tractography and found similar trends (S9B Fig in SI). We also corroborate our findings with publicly available probabilistic tractography data from a recent study on HCP Young Adult (S9C Fig in SI).

Notably, when we further evaluate the commute times based on the top principal component of commute time matrix in addition to that of FCs, the correlation obtained for 434 UK Biobank individuals further increased to $0.36 \pm 0.14$ (Fig 3C, *right violin plot*). Likewise, the correlation increased to $0.32 \pm 0.12$ (S9A Fig in SI, *rightmost violin plot*) when repeating the analysis for the HCP Young Adult dataset. The 33–38% increase in correlation (from 0.26 to 0.36 for UK Biobank, and 0.24 to 0.32 for HCP Young Adult) upon considering the principal components of both commute times and FCs supports the view that the global features of structure make a concrete contribution to shaping the dominant functional couplings reflected by FCs. In other words, the correlation between structure and function, which is marginal but statistically significant becomes more apparent by removing noise from possible measurement errors. Taking the top mode of the commute time matrix is not helpful for the simulated data in the previous section, because the inputted structure in the simulation exactly corresponds to the outputted function (S10 Fig in SI).

Throughout the main text, commute time is calculated based on the number of tracts between brain regions. However, we could adopt alternative adjacency matrices in Equation 6 to calculate the commute times based on different perspectives of brain network topology. In S11 Fig in SI, we present the commute time results when calculated with a binary representation of the adjacency matrix (*blue distributions*) or a weighted connectivity matrix based on tract lengths (*green distributions*). Commute time calculated based on the number of tracts, also called density (*orange distributions*), outperforms both of these alternative adjacency definitions. A possible explanation for the relatively lower performance of tract lengths compared to densities could be the adjustment of conduction velocities of longer tracts such that signals reach their destinations at similar times regardless of their length [48].

An alternative mathematical approach to calculate commute time is to express it in terms of mean first passage time, rather than hitting time. Mean first passage time is the large time limit approximation of hitting time (S2 Text in SI). A previous study found mean first passage time to be one of the best FC correlates among 40 structure-based predictors [22]. We also find that commute time as calculated by mean first passage time does just as well as when calculated by hitting time (S12A Fig in SI) across UK Biobank individuals, because the two diffusion-based metrics are highly correlated (S12B Fig in SI).

### Structure-function coupling strength weakly relates to biomarkers

Commute time outperforms other metrics at predicting function on average, however, the standard deviation is elevated compared to other communication metrics (Fig 3C and S9A Fig in SI): there is a large variation across individuals with correlations ranging from −0.059 to 0.58. The range further increases to –0.08 to 0.70 when focusing on FCs' and

commute times' first principal components for the UK Biobank (Fig 3C, rightmost violin plot). We investigate whether these variations could be attributed to specific subgroups of individuals, e.g., whether the commute time-FC correlations are systematically weaker for individuals with mental health or nerve disorders, as defined by the International Classification of Diseases-10 codes F and G, respectively [49]. S4 Table in SI lists the names of the disease present within our UK Biobank subset. In Fig 4A, we found that UK Biobank individuals with mental disorders have slightly weaker commute time-FC correlations compared to healthy individuals, but those results are not statistically significant (KS = 0.55, p = 0.12 (S5 Table in SI)) because of the sample size (only 4 individuals with mental disorders). Interestingly, individuals diagnosed with nerve disorders display structure-function correlations similar to those of healthy individuals. These results are surprising because mental health disorders are normally diagnosed by behavior, while neurological diseases have a clearer connection to brain structure when directly affecting the central nervous system. For example, a stroke leads to neuronal loss which reroutes signal transmission [5,50,51]. However, our preliminary results indicate that mental health disorders may be just as vulnerable to impaired structure-function relationships, if not more. When looking in more detail

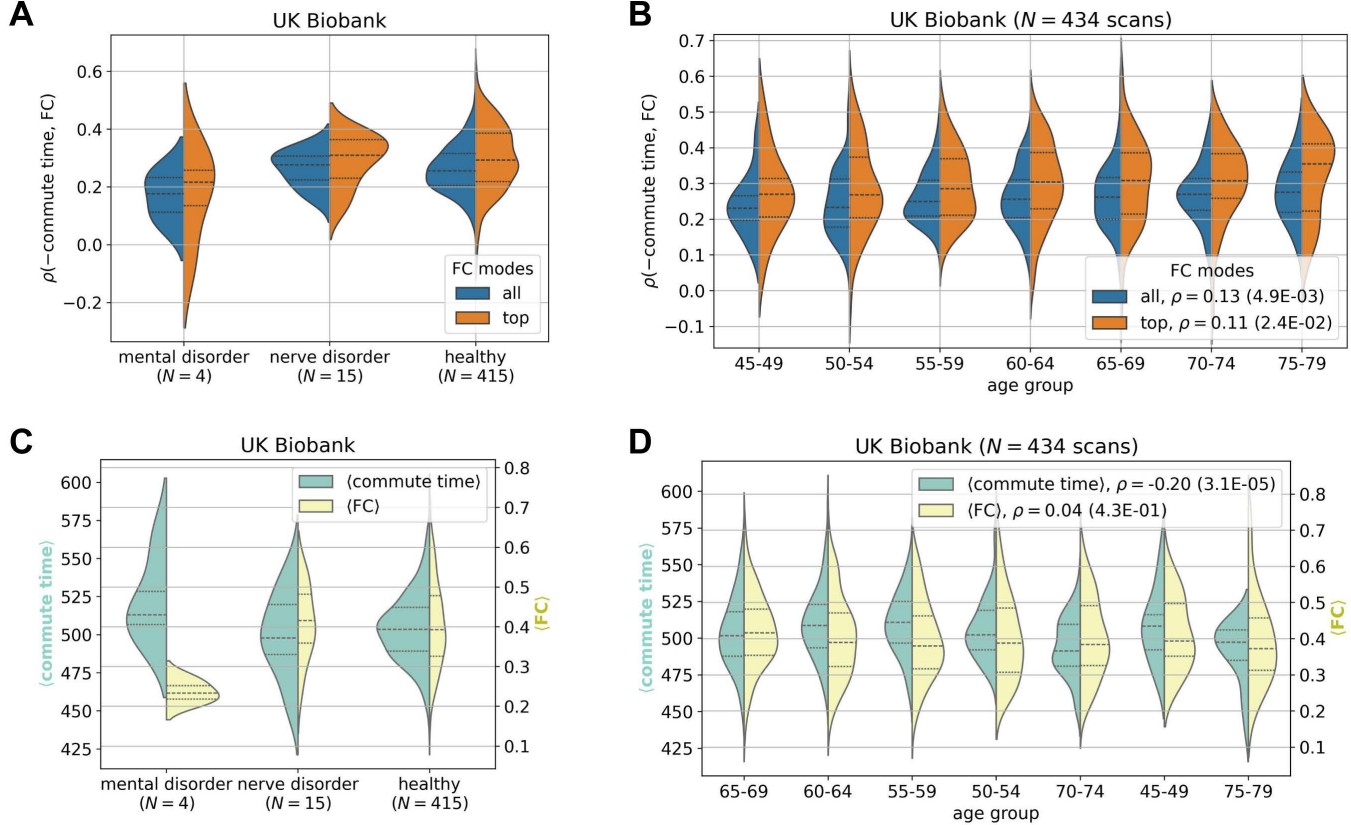

**Fig 4. The strength of the commute time-functional connectivity relationship demonstrates weak pathological significance and weak dependency on age.** fMRI and dMRI data are processed according to the Desikan-Killiany atlas throughout (84 brain regions). **A.** Small differences are seen between individuals with mental and nerve disorders compared to healthy individuals in the UK Biobank. The corresponding Kolmogorov-Smirnov pairwise tests are not significant (S5 Table in SI). **B.** Older brains have slightly stronger correlations between commute time and FC. Two distributions are plotted, where the orange distribution only includes the top mode of the FC matrix. **C-D.** Corresponding plots of average commute time (green) and average FC (yellow) across all possible pairs distributed over individuals. **C.** Individuals with mental health disorders have slightly higher average commute times and much lower average FCs. **D.** Strengthening commute time-FC correlations shown in **B** seem to be driven by a decrease in average commute time across age. Note that all reported correlations are calculated by considering all data points, i.e., they are not calculated for the binned data.

into average commute time and FC values across brain pairs, we find that individuals with mental disorders exhibit the greatest deviation from healthy individuals, especially their average FC values (Fig 4C). The corresponding slightly longer commute times also point to slower communication between brain regions.

We proceed to test whether aging results in a change in the relationship between brain structure and function. This has been shown to be the case with other metrics [22] and might be expected because older brains have been inferred to have lower connection strengths between regions [34]. In Fig 4B, we surprisingly find that the magnitude of the commute time-FC correlation slightly increases with age. In other words, older brains have a stronger structure-function relationship. Our result departs from the recent finding that among 40 different predictors (which do not include commute time), the correlation of the best predictor for a given individual decreases with age [22]. This contrasting result highlights the importance of metric choice in making inferences on brain-wide aging trends. This trend cannot be perceived however, when HCP Young Adult dataset is considered (S13B Fig in SI). The lack of any aging trends in commute time-FC correlations in this dataset may be due to the narrow range compared to the UK Biobank, 22–37 compared to 46–79 years old (S8 Table in SI). The aging trend in the UK Biobank seems to be driven by overall decreases in commute time as age increases (Fig 4D). This surprising result can be reasoned from our recent report demonstrating that older individuals have more white matter tracts on average when performing tractography with more advanced and accurate methods [34]. Trivially, if more edges are randomly placed in a network, the commute time decreases.

To further test the impact of our results on behavior, we analyzed Mini-Mental State Examination scores collected by the HCP Young Adult dataset and found minimal differences between cogitative ability and $\rho$(commute time, FC) (S13A Fig in SI). Finally, as a control, we confirm that sex does not confound the relationship between commute time and FC (S14 Fig in SI).

## Discussion

Driven by recent successes of communication metrics at capturing brain dynamics, we investigate whether commute times calculated from networks derived from dMRI could capture the functional connectivities derived from fMRI. We demonstrate that commute time outperforms other metrics such as communicability and search information. This improved performance is presumably because commute time stochastically captures signal propagation based on the entire connectome topology, rather than the local connectivity only. The significance of considering global properties is even more apparent when we extract the principal components of both commute times and functional connectivities, resulting in 33–38% increases in correlations. Although other communication metrics aim to also capture the entire connectome topology, only commute time is rooted in a physical model of neurosignaling based on Markovian dynamics. This grants commute time several advantageous properties that neither communicability nor search information simultaneously both have. As with communicability, commute time has an analytical expression. As with search information, commute time can distinguish between pairs of regions with no tracts. All three structure-based metrics have no fitted parameters. Commute time is mathematically calculated based on the pseudoinverse of the Laplacian matrix. Thus, commute time can be classified as a communication metric that incorporates structural harmonics, another independent approach also used to link brain structure and function [52–56].

Results investigating the pathological implications of weak commute time-FC correlations are inconclusive because of small sample sizes. We only have access to 434 UK Biobank individuals, 4 and 15 of which have been diagnosed with a mental health disorder and nerve disorder, respectively. In future studies, we will have access to more UK Biobank dMRI and fMRI scans to more thoroughly investigate the intersection between brain structure and function with disease. Intriguingly, we already find that older adults have slightly stronger structure-function relationships than younger adults. White matter is one of the last brain areas to fully develop in young adulthood [57] and our results suggest that its continued maturation in advanced ages could lead to a stronger coupling with function.

Many studies focus on intra-hemisphere white matter tracts in their data analysis [18,19], because dMRI is known to be less accurate in reconstructing longer tracts [58,59]. In view of these considerations, we repeated our analysis for intra-hemisphere signaling. The resulting average correlation coefficient increased from $-0.26 \pm 0.08$ (entire brain) to $-0.31 \pm 0.11$ (left-hemisphere only) across the 434 UK Biobank individuals (S15 Fig in SI). Notably, commute time still better correlates with the intra-hemisphere FCs than the three other metrics. The improvement in correlation magnitude compared to that obtained with communicability is 35% ($\rho = 0.23$).

Despite improved structure-function coupling, the overall correlation between commute times based on dMRI data and FCs functional correlations (FCs) based on fMRI remain bounded to $\leq 0.38$ even after 'denoising' by considering the top eigenmodes of both quantities. This is in contrast to the correlation we observe between the same commute times and FCs obtained from simulations of brain signaling on a dMRI-derived network topology (Fig 2). One clear reason for this discrepancy are the long commute times predicted between regions across the two hemispheres of the brain, when their FCs from fMRI are not weak. Structural connectivity matrices are sparse, while FC matrices are dense [11,16]. Commute time helps transform structural connectivity matrices to become denser; however, it still pales in comparison to FC matrices (Fig 1).

Particularly noticeable are the strong homotopic connections (diagonal elements of the two off-diagonal quadrants in (Fig 1) present in FC, but not in the structural connectivity or commute time matrices. Homotopic connections correspond to interhemispheric connections between the same functional brain region across the two hemispheres and have been previously shown to be some of the most stable functional connections [16,45,60,61]. In an attempt to better capture FC, we manually introduce homotopic tracts in the adjacency matrices. This leads to marginal improvements in the commute time-FC correlation when 100 tracts are added to each possible homotopic pair for individuals in the HCP Young Adult dataset ($-0.30 \pm 0.08$ vs $-0.24 \pm 0.06$ as shown in S16A Fig in SI vs S9A Fig in SI. We examine the correlation strength as a function of the number of added homotopic tracts and find that improvements level off at around 1,000 added tracts. Compared to randomly added tracts, homotopic added tracts considerably improve commute time-FC correlations (S16B-C Fig in SI).

The mathematical expression for commute time is based on a first order Markovian process, i.e., it only considers the current node that it is located in when randomly traveling from a source node to a destination node. First order Markovian processes ignore memory effects [62], which may not be the case in brain signaling. However, we numerically find that calculating commute time based on a second-order Markov process where transitions are not allowed to immediately backtrack strongly correlates with a first order Markov process for an arbitrary brain structure (S17A Fig in SI). Therefore, the success of the commute time expression in relating to FC values from fMRI is not necessarily limited by its inability to capture memory effects for brain structures because minimal information is added when considering second order and even third order memory effects (S17 Fig in SI).

The observed weak but robust commute time-FC correlations provide evidence that brain regional activity at resting-state approximately diffuse along its network topology to reach its destination. By outperforming other communication measures, our work identifies a dominant feature of brain organization. Machine learning methods trained to predict FC based on structural connectivity have obtained stronger correlations with FC across individuals, with Pearson correlations greater than 0.5 [63–68]. However, reasons from underlying success remain difficult to elucidate and separate from simply extensive parameter fitting. Our physics-based approach provides a benchmark that the next generation of communication models must surpass to advance our understanding of the brain, with correlation values ($\rho = 0.36 \pm 0.14$ across 434 UK Biobank individuals) that begin to reach those of machine learning approaches.

Finally, we must consider whether non-neuronal factors drive fMRI signals [69–73]. We use a hemodynamic response function to output a BOLD-like signal in our simulations of brain function. However, non-neuronal neurovascular variability is not considered and could undermine the structure-function relationship, further explaining the observed weak commute time-FC correlation for fMRI data. Indeed, when applying a well-established point process method to first deconvolve the

 

hemodynamic response function from the BOLD signal [74,75] and then calculating FC, we find slightly stronger commute time-FC correlations for the UK Biobank (S18 Fig in SI vs Fig 3C). Incorporating other factors to generate the connectivity matrix to calculate commute time, such as neurotransmitter gene expression [76–78] and connection directionality [79,80], are also important future considerations to further enhance our understanding of the forces underlying FC as measured by fMRI. The importance of extrasynaptic neurotransmission, and in particular the modulatory effects of monoaminergic and neuropeptidergic signaling on the synaptic connectome, has been highlighted in multilayered mappings of the *C. elegans* connectome [81,82]. These studies illustrate the rich environment underlying neurosignaling beyond axonal transmission. The universal theme in biology that structure determines function may require a more encompassing definition of 'structure' to meaningfully apply to the brain.

## Materials and Methods

### Commute time

We expand upon the main text's derivation for $H_{ij}$ by filling in the gaps between Equation 2–5. We can simplify Equation 2 by noting that the sum of the transition matrix $M$ for any row equals to 1.

$$H_{ij} = \sum_{k}^{N} M_{ik} + \sum_{k,\ k\neq j}^{N} M_{ik}H_{kj}$$

$$= 1 + \sum_{k,\ k\neq j}^{N} M_{ik}H_{kj} \tag{7}$$

Next, we write Equation 7 in matrix notation by defining matrices with a ^ symbol, designating that the $j^{th}$ element from row vectors and $j^{th}$ columns and rows from $M$ are removed from consideration.

$$\hat{H}_j = \hat{1}^{T} + \hat{M}\hat{H}_j \tag{8}$$

Note that $\hat{H}_j$ is a 1 x (N-1) row vector and $\hat{M}$ is a (N-1) x (N-1) matrix. We can simplify the expression and solve for $\hat{H}_j$ by expanding the transition matrix $\hat{M}$ in terms of the Laplacian matrix $\Gamma = D - A$.

$$\hat{H}_j = \hat{1}^{T} + \hat{D}^{-1}\hat{A}\hat{H}_j$$

$$\hat{H}_j = \hat{1}^{T} + \hat{D}^{-1}\left(\hat{D} - \hat{\Gamma}\right)\hat{H}_j$$

$$\hat{H}_j = \hat{1}^{T} + \hat{H}_j - \hat{D}^{-1}\hat{\Gamma}\hat{H}_j$$

$$\rightarrow \hat{H}_j = \hat{\Gamma}^{-1}\hat{D}\hat{1}^{T} \tag{9}$$

The final form of Equation 9 can be rewritten in terms of components as,

$$H_{ij} = \sum_{k,k\neq j}^{N} \left[\hat{\Gamma}^{-1}\right]_{ik} \hat{D}_{kk} \tag{10}$$

In the Supplement, we demonstrate how we can express $\hat{\Gamma}^{-1}$ in terms of $\Gamma^{-1}$ by sequentially adding one row and then one column to $\hat{\Gamma}^{-1}$ (Equation 15). We utilize Greville's formula twice to calculate the updated matrix inverses [83,84]. Below is the result,

$$H_{ij} = \sum_{k,k\neq j}^{N} \left( \left[\Gamma^{-1}\right]_{ik} - \left[\Gamma^{-1}\right]_{ji} - \left[\Gamma^{-1}\right]_{jk} + \left[\Gamma^{-1}\right]_{jj} \right) \hat{D}_{kk}$$

$$= \sum_{k}^{N} \left( \left[\Gamma^{-1}\right]_{ik} - \left[\Gamma^{-1}\right]_{ji} - \left[\Gamma^{-1}\right]_{jk} + \left[\Gamma^{-1}\right]_{jj} \right) D_{kk}$$

(11)

The final expression in Equation 11 is equivalent to Equation 5 in the main text. Commute time can then be obtained by summing $H_{ij}$ and $H_{ji}$ (Equation 6).

Commute time ($C_{ij}$) is closely related to other physical properties besides random walks. In the elastic network model (ENM) of proteins [85,86], the mean-square fluctuations ($\langle\Delta\boldsymbol{R_{ij}} \cdot \Delta\boldsymbol{R_{ij}}\rangle$) in the distance vectors $\boldsymbol{R_{ij}}$ between nodes $i$ and $j$, are proportional to commute time between these nodes [23,31]. Note that in the ENM representation of proteins, the nodes describe the amino acids, and the springs, their interactions/couplings. The resistance distance ($\Omega_{ij}$) between two nodes in an electrical network is also proportional to commute time [25,87,88]. $\langle\Delta\boldsymbol{R_{ij}} \cdot \Delta\boldsymbol{R_{ij}}\rangle$ and $\Omega_{ij}$ have independent derivations based in statistical mechanics and electromagnetism, respectively.

$$\langle\Delta\boldsymbol{R_{ij}} \cdot \Delta\boldsymbol{R_{ij}}\rangle \propto \Omega_{ij} \propto C_{ij} \propto \left[\Gamma^{-1}\right]_{ii} + \left[\Gamma^{-1}\right]_{jj} - 2\left[\Gamma^{-1}\right]_{ij}$$

(12)

Commute time is the most relevant property in the current application because its basis in the Markovian propagation of signals provides a physical mechanism for brain signaling.

**Communicability**

Communicability (CMY) is analytically expressed in terms of the graph's weighted adjacency matrix (**A**) [89,44,90].

$$CMY(i,j) = \sum_{n=0}^{\infty} \frac{1}{n!} \left( \frac{A_{ij}}{\sqrt{D_{ii}D_{jj}}} \right)^n = \exp\left[ \frac{A_{ij}}{\sqrt{D_{ii}D_{jj}}} \right]$$

(13)

The weighted adjacency matrix is normalized based on the total number of tracts for a node $i$, $D_{ii}$ (Equation 3). By scaling by the factorial of the path length, the summation can be exactly expressed as an exponential. This factorial factor can physically correspond to springs on a surface plane [89]. However, it is unclear how the physical movement of springs in a planar space corresponds to brain signaling along its structure.

**Search information**

Search information (SI) is the amount of information in bits that a random walk needs in order to follow the shortest path based on tract lengths [19,91]. It is expressed in terms of the transition matrix **M**, which can be computed in terms of the weighted adjacency matrix as noted in the main text when deriving commute time (Equation 3).

$$SI(i,j) = \log_2\left[M_{ia}M_{ab}M_{bc} * \ldots * M_{zj}\right]$$

(14)

The nodes *a, b, …, z* are located on the shortest path connecting node *i* to node *j*. We calculate search information using previously published code in the Brain Connectivity Toolbox [92]. ChatGPT was used to help translate the original Matlab code into Python.

## Mean-field Ising simulation of brain function

The following steps are performed to generate the functional connectivity (FC) matrix from the Ising model.

1. Set the static structure based on an individual's dMRI data. Edges are assigned weights $A_{ij}$ corresponding to the number of white matter tracts observed between pairs of regions in diffusion MRI.

2. Set the global coupling strength λ for all edges. λ is the only parameter in the mean-field Ising simulation and dictates the energy of the system, $E = -\lambda \sum_{ij} A_{ij} s_i s_j$, where $s_i$ corresponds to the spin state of region *i* and can either be +1 or −1. The larger the λ, the more likely that all nodes will be all up or all down. For smaller λ, entropy wins out and individual spins are equally likely to be up or down regardless of the states of their connected neighbors.

3. Initialize the simulation by randomly assigning spins of +1 or −1 to each of the nodes.

4. Evolve the simulation by one timestep by randomly attempting to flip 15% of the nodes' spins. We accept attempted sets of spin flips according to the Metropolis-Hastings algorithm [93], which more likely accepts spin flips that lower the system's energy while obeying detailed balance. The exact number of randomly attempted spin flips for a certain number of iterations does not matter so long as the simulation is in the equilibrium regime.

5. Repeat step 4) for a total of 5000 times to ensure that the state space is well-sampled.

6. Convolute the Ising spins with the Volterra kernel hemodynamic response function [42] to output a fMRI BOLD-like signal.

7. Correlate the entire time-series across pairs of regions to create the FC matrix.

Fig 5 shows a schematic of the Ising simulation for a system with seven nodes and one spin flip attempted per timestep.

## dMRI processing

We take preprocessed dMRI scans from the UK Biobank [94] and calculate connectivity matrices using the Diffusion Imaging in Python software [30]. For each individual, we determine the number of streamline counts of white matter tracts between regions and their average lengths. We input the Desikan-Killiany atlas [29,30] to distinguish between white

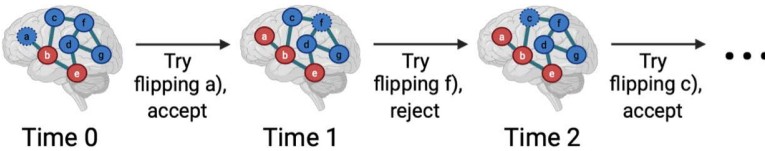

**Fig 5. Schematic of the mean-field Ising simulation evolving in time.** We start with a fixed structure whose spins are randomly initialized either up (blue) or down (red) at time 0. We then randomly choose a node and attempt to flip its spin. At time 0, we attempt to flip node **a)**. It is accepted according to the Metropolis-Hastings algorithm because the energy becomes stabilized by λ units. At time 1, we attempt to flip node **f)**. Because the energy is destabilized by 3*λ, we accept it with a probability of $e^{-3\lambda}$ according to the Metropolis-Hastings algorithm. In the figure, this particular realization of **f)**'s spin flip is rejected. At time 2, we attempt to flip node **c)**. The change in energy is 0, so we accept the spin flip. This schematic is simplified such that all edges have the same weight. The simulation also accounts for the number of tracts for each edge derived from the diffusion MRI of an arbitrarily chosen individual. Figure created with biorender.com.

matter and neuronal regions in the gray matter. We perform deterministic tractography and reconstruct the orientation distribution function using the Constant Solid Angle (Q-Ball) method with a spherical harmonic order of 6 [95]. The relative peak threshold is set to 0.8 with a minimum separation angle of $45°$. We only seed voxels in the white matter and count tracts that end in the gray matter. Minimum step size of tracts is 0.5 mm. Self-loops are removed from consideration such that all diagonal elements in the structural connectivity matrix are equal to 0.

To make sure results are not an artefact of the particular brain atlas, we also process dMRI scans according to the Talairach atlas [43]. Regions that have zero inferred white matter tracts are removed from consideration. For some individuals, different regions are removed from consideration. The average number of nodes per individual is 727 [96].

To make sure results are not an artefact of the tractography method, we also perform probabilistic tractography and reconstruct the orientation distribution function using the same Constant Solid Angle (Q-Ball) method with a spherical harmonic order of 6.

The same procedure is applied to the HCP Young Adult dataset from accessed preprocessed dMRI [97,98] to generate the structural connectivity matrices. Demographic details for both the HCP Young Adult and UK Biobank dataset can be found in S7 Table in SI.

### fMRI processing

We take preprocessed resting-state fMRI scans from the UK Biobank [99]. The cleaned, voxel space time series are band-pass filtered to only include neuronal frequencies (0.01 to 0.1 Hz) and smoothed at a full width at half maximum of 5 mm. Finally, we parcellate into 84 brain regions of interest according to the Desikan-Killiany atlas [29,30], the same as in dMRI processing. We also process fMRI scans according to the Talairach atlas [43] and remove from consideration those brain regions that have no white matter tracts (see dMRI processing). Functional connectivity is calculated by correlating the entire processed time-series between brain regions according to the Pearson correlation coefficient.

The same procedure Is applied to the HCP Young Adult dataset from accessed preprocessed fMRI [97] to generate the functional connectivity matrices. Acquisition details such as field strength and repetition time for both datasets can be found in S8 Table in SI.

### Supporting information

**S1 File.** A total of 18 figures, 9 tables, and 2 texts are included to support the conclusions in the main text. References [100–103] are cited in the Supplementary Information.
(PDF)

### Acknowledgments

We would like to thank Botond Antal for sharing his fMRI expertise and sharing his Wilson-Cowan Python function wrappers on the Virtual Brain and Mobolaji Williams for theoretical support on deriving the analytical expression for commute time. We would also like to thank Professor Daniele Marinazzo for his insightful comments. This research has been conducted using the UK Biobank Resource under Application Number 37462. Data were provided in part by the Human Connectome Project, WU-Minn Consortium (Principal Investigators: David Van Essen and Kamil Ugurbil; 1U54MH091657) funded by the 16 NIH Institutes and Centers that support the NIH Blueprint for Neuroscience Research; and by the McDonnell Center for Systems Neuroscience at Washington University.

### Author contributions

**Conceptualization:** Rostam Razban, Ivet Bahar.

**Data curation:** Rostam Razban.

**Formal analysis:** Rostam Razban, Anupam Banerjee.

**Funding acquisition:** Ivet Bahar.

**Investigation:** Rostam Razban, Anupam Banerjee, Lilianne R. Mujica-Parodi, Ivet Bahar.

**Supervision:** Ivet Bahar.

**Writing – original draft:** Rostam Razban, Ivet Bahar.

**Writing – review & editing:** Rostam Razban, Anupam Banerjee, Lilianne R. Mujica-Parodi, Ivet Bahar.

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
