## [Decision Letter · Decision Letter 0]

21 May 2025

PONE-D-25-16428The role of structural connectivity on brain function through a Markov model of signal transmissionPLOS ONE

Dear Dr. Razban,

Thank you for submitting your manuscript to PLOS ONE. After careful consideration, we feel that it has merit but does not fully meet PLOS ONE’s publication criteria as it currently stands. Therefore, we invite you to submit a revised version of the manuscript that addresses the points raised during the review process.

We look forward to receiving your revised manuscript.

Kind regards,

Alpen Ortug, PhD

Academic Editor

PLOS ONE

Journal Requirements:

2. Thank you for stating the following financial disclosure: [Support from the National Institutes of Health (R01 GM139297) is gratefully acknowledged by IB.]. 

Reviewers' comments:

Reviewer's Responses to Questions

**Comments to the Author**

1. Is the manuscript technically sound, and do the data support the conclusions?

Reviewer #1: Yes

Reviewer #2: Yes

2. Has the statistical analysis been performed appropriately and rigorously? 

Reviewer #1: Yes

Reviewer #2: Yes

3. Have the authors made all data underlying the findings in their manuscript fully available?

Reviewer #1: Yes

Reviewer #2: No

4. Is the manuscript presented in an intelligible fashion and written in standard English?

Reviewer #1: Yes

Reviewer #2: Yes

5. Review Comments to the Author

Reviewer #1: I am happy to recommend this for publication with minor revisions. The manuscript presents an interesting and novel metric that aims to relate structural and functional brain connectivity by modelling brain signal propagation as a Markovian process, using commute time. This is a well-motivated study that contributes meaningfully to the ongoing effort to reconcile structural and functional connectivity in the brain.

STRENGTHS:

The inclusion of results from multiple brain atlases demonstrates the robustness of the metric, and adds to the reproducibility and generalisability of the work. The comparison with existing metrics, and exploration of age- and health-related differences provides some useful insights.

RECOMMENDATIONS:

Physical vs. topological distance: It was not immediately clear from the text whether the commute time is computed using physical distance or graph-theoretic distance (i.e. number of edges traversed). This is an important distinction. For instance, a long-range callosal axon connecting opposite hemispheres likely differs substantially in transmission properties from a short-range local tract, even if both are represented by a single edge in the connectivity graph. I encourage the authors to clarify this point and, if applicable, to discuss the implications of using graph distance versus physical distance, especially in light of the heterogeneity in axon length, conduction velocity, and energy cost.

Consideration of extrasynaptic communication: One possible reason why structural connectivity often fails to fully explain functional connectivity is that communication in the brain is not limited to direct axonal or synaptic transmission. There is growing evidence for extrasynaptic or volume transmission via neuromodulatory systems such as monoamines and neuropeptides. These systems can exert widespread influence, bypassing classical synaptic networks. Although difficult to quantify in current human neuroimaging, this possibility should be acknowledged. The authors are encouraged to discuss such alternative modes of signalling and their potential to account for observed discrepancies between structural and functional connectivity. Recommended references:

Bentley et al. (2016), The multilayer connectome of C. elegans, PLOS Computational Biology. https://doi.org/10.1371/journal.pcbi.1005283

Ripoll-Sánchez et al. (2023), The neuropeptidergic connectome of C. elegans. Neuron. https://doi.org/10.1016/j.neuron.2023.09.043

While these studies are in C. elegans, they exemplify the importance of considering multiple modes of communication in connectomic analyses.

Line 66–70: Clarification of communicability. The brief description of communicability may leave some readers unfamiliar with the metric uncertain about its mechanics and implications. Please elaborate.

Line 483 onwards: Notation. The notation used for mean-square fluctuations appears to use less-than/greater-than symbols, which could be misinterpreted as relational operators. If the authors intended to denote averaging, they should replace these with angled brackets, as is conventional (e.g., 〈·〉).

Reviewer #2: The current paper uses Markov models to examine the relationship between structure and function in human brain networks. It specifically uses commute time as a measure to quantify signal transmission between brain regions. The paper is well written and provides a thorough description of the analysis and discussion of results. I have a few comments as follows:

1) The introduction provides useful information for the study. However, more information on the motivation for using the commute time measure, its biological plausibility, and comparison with other measures of signal transmission, including neural mass models, should be provided.

2) The rationale for choosing only two communication measures (communicability and search information) was not clear. Other communication measures, such as mean first passage time, which are based on the diffusion process, should be considered for comparisons.

3) The analysis based on simulated FC was informative. However, have the authors considered including randomized FC/null models to check whether the observed relationships in simulated and actual FC are meaningfully constrained by the actual network architecture.

4) I may have missed this, but it seems that the statistics for correlation between the simulated functional connectivity and commute time are only reported at the individual level. It would strengthen the analysis to also provide population level statistics (e.g., group means, statistical significance).

5) The final section of the results, “Structure-function coupling strength weakly relates to biomarkers,” should be expanded. Specifically, it would be helpful to specify which brain diseases and mental disorders are included in the analysis. For each condition, more detailed information should be provided on how commute time is disrupted, such as the regional changes of the number of streamlines in disease condition and their correlation with the commute time. Additionally, differences in functional and structural connectivity between disease and healthy groups should be reported to have a better context for alterations of structure-function coupling.

6. PLOS authors have the option to publish the peer review history of their article (what does this mean? ). If published, this will include your full peer review and any attached files.

**Do you want your identity to be public for this peer review?** For information about this choice, including consent withdrawal, please see our Privacy Policy .

Reviewer #1: No

Reviewer #2: No

---

## [Author Response · Author response to Decision Letter 1]

21 Jul 2025

Please see the Response_to_Reviewers.doc file.

---

## [Decision Letter · Decision Letter 1]

12 Aug 2025

The role of structural connectivity on brain function through a Markov model of signal transmission

PONE-D-25-16428R1

Dear Dr. Razban,

We’re pleased to inform you that your manuscript has been judged scientifically suitable for publication and will be formally accepted for publication once it meets all outstanding technical requirements.

Kind regards,

Alpen Ortug, PhD

Academic Editor

PLOS ONE

Additional Editor Comments (optional):

Reviewers' comments:

Reviewer's Responses to Questions

**Comments to the Author**

1. If the authors have adequately addressed your comments raised in a previous round of review and you feel that this manuscript is now acceptable for publication, you may indicate that here to bypass the “Comments to the Author” section, enter your conflict of interest statement in the “Confidential to Editor” section, and submit your "Accept" recommendation.

Reviewer #1: All comments have been addressed

Reviewer #2: All comments have been addressed

2. Is the manuscript technically sound, and do the data support the conclusions?

Reviewer #1: Yes

Reviewer #2: Yes

3. Has the statistical analysis been performed appropriately and rigorously? 

Reviewer #1: Yes

Reviewer #2: Yes

4. Have the authors made all data underlying the findings in their manuscript fully available?

Reviewer #1: Yes

Reviewer #2: No

5. Is the manuscript presented in an intelligible fashion and written in standard English?

Reviewer #1: Yes

Reviewer #2: Yes

6. Review Comments to the Author

Reviewer #1: Thank you for taking the time to edit the manuscript. The recent revisions addressed all of my concerns, and have greatly improved the manuscript. I am happy to recommend this for publication.

Reviewer #2: (No Response)

7. PLOS authors have the option to publish the peer review history of their article (what does this mean? ). If published, this will include your full peer review and any attached files.

**Do you want your identity to be public for this peer review?** For information about this choice, including consent withdrawal, please see our Privacy Policy .

Reviewer #1: No

Reviewer #2: No

---

## [Editor Report · Acceptance letter]

PONE-D-25-16428R1

PLOS ONE

Dear Dr. Razban,

I'm pleased to inform you that your manuscript has been deemed suitable for publication in PLOS ONE. Congratulations! Your manuscript is now being handed over to our production team.

Kind regards,

on behalf of

Dr. Alpen Ortug

Academic Editor

PLOS ONE